# Transformation and Cytotoxicity of Surface-Modified Silver Nanoparticles Undergoing Long-Term Aging

**DOI:** 10.3390/nano10112255

**Published:** 2020-11-13

**Authors:** Chengfang Pang, Panhong Zhang, Yunsong Mu, Jingzheng Ren, Bin Zhao

**Affiliations:** 1The Intelligent Drug Delivery and Sensing Using Microcontainers and Nanomechanics (IDUN), Department of Health Technology, Technical University of Denmark, Ørsteds Plads 345C, 2800 Kongens Lyngby, Denmark; 2State Key Laboratory of Environmental Chemistry and Ecotoxicology, Research Center for Eco-Environmental Sciences, Chinese Academy of Sciences, Beijing 100085, China; panhongzhang2015@yeah.net (P.Z.); binzhao@rcees.ac.cn (B.Z.); 3School of Environment and Natural Resources, Renmin University of China, Beijing 100872, China; muyunsong323@163.com; 4Department of Industrial and Systems Engineering, Hong Kong Polytechnic University (PolyU), Hung Hom, Kowloon, Hong Kong, China; jingzheng.jz.ren@polyu.edu.hk

**Keywords:** nanoparticle transformation, aging, toxicity, silver nanoparticles, surface-modified nanoparticles

## Abstract

Silver nanoparticles (AgNPs) are constituents of many consumer products, but the future of their production depends on ensuring safety. The stability of AgNPs in various physiological solutions and aging in storage may affect the accuracy of predicted nanoparticle toxicity. The goal of this study was to simulate the transformation of AgNPs in different media representatives to the life cycle in the environment and to identify their toxicity to Hepa1c1c7 cells in a long-term aging process. AgNPs coated with citrate, polyethylene glycol (PEG), polyvinylpyrrolidone (PVP), and branched polyethyleneimine (BPEI) were studied. Our results show that the exposure media had a significant impact on the transformation of AgNPs. Citrate-coated AgNPs showed significant aggregation in phosphate-buffered saline. The aging of AgNPs in optimal storage showed that the charge-stabilized particles (citrate) were more unstable, with significant aggregation and shape changes, than sterically stabilized particles (PEG AgNPs, PVP AgNPs). The BPEI AgNPs showed the highest dissolution of AgNPs, which induced significantly increased toxicity to Hepa1c1c7 cells. Overall, our findings showed that storage and media of AgNPs influenced the transformation of AgNPs and that the resulting changes in the AgNPs’ physicochemical properties influenced their toxicity. Our study contributes to the understanding of AgNPs’ transformations under realistic exposure scenarios and increasing the predictability of risk assessments.

## 1. Introduction

Silver nanoparticles (AgNPs) are one of the most commonly used engineered nanomaterials in many applications (e.g., catalysis, antimicrobial agents, medical device coatings, drug-delivery formulations, water treatment, and anticancer agents), which results in their significant release into the environment and induces potential risks to ecological receptors and humans [1,2]. While the body of work studying the implications of AgNPs is growing, gaps in our understanding of AgNPs’ health and safety still exist. Specifically, long-term transformations that AgNPs undergo in the environment are of significant concern.

The high surface to volume ratio of AgNPs makes them highly dynamic in natural systems and significantly changes their physical and chemical properties in the environment, including their ability to oxidize and sulfidize. Sulfidation of the nanoparticles can change their aggregation state, surface chemistry, and charge, as well as their ability to release toxic Ag ions. In turn, this modifies their persistence and toxicity [3,4]. Additionally, AgNPs form a (bio)molecule corona, which dramatically changes their physical properties (e.g., aggregation, deposition) and ultimate toxicity [5,6]. Consequently, systematic studies of complex transformation of AgNPs in biological and environmental systems, and their toxicity mechanisms and resulting toxicity under realistic exposure conditions, are urgently required.

Moreover, complex nano–bio interactions on the surface of AgNPs with different coatings can also cause toxicity. Different surface coatings are used to develop faster electronics, brighter displays, and more sensitive diagnostic agents for medical imaging. Previous studies have demonstrated that the toxicity of nanomaterials’ particle size, composition, surface charge, shape, and aggregation drive the toxicity of nanomaterials [7,8,9,10,11]. Therefore, a small change in these properties may dramatically influence AgNPs’ physiological response. Gliga et al. showed that AgNPs’ toxicity in human lung cells is associated with the rate of intracellular Ag release, which depends on the size of the AgNPs [9]. Apart from the size of nanoparticles, in vivo studies have shown that AgNPs’ surface chemistry is largely responsible for their fate because of adsorption of serum ingredients, mainly proteins [12,13]. For example, Pang et al. detected intracellular localization of positive surface-charged, branched, polyethyleneimine-coated AgNPs in the nucleus of cells, which demonstrated that the toxicity of AgNPs depends on the surface charge [11]. Even though these studies have shown that the toxicity of AgNPs is relevant to their surface coatings, how the surface coating can change the transformation of AgNPs in environmental and biological systems is unknown. This is crucial to identify realistic AgNP exposure and hazards to the environment and human health.

To address these knowledge gaps, we investigated the transformation of different surface-coated AgNPs in different media and identified the toxicity of transformed AgNPs to Hepa1c1c7 cells in a long-term aging process. The Hepa1c1c7 test is applied as a tool to gain an insight into the relative development of toxicity during the different treatments. Four types of AgNPs coated with citrate, polyethylene glycol, polyvinylpyrrolidone, and branched polyethyleneimine (Citrate AgNPs, PEG AgNPs, PVP AgNPs, and BPEI AgNPs, respectively) were tested in the study.

## 2. Materials and Methods

### 2.1. Characterization of AgNPs

Citrate, polyethylene glycol, polyvinylpyrrolidone, and branched polyethyleneimine (Citrate AgNPs, PEG AgNPs, PVP AgNPs, and BPEI AgNPs, respectively) were used as coatings for AgNPs. These materials were purchased from nanoComposix (San Diego, CA, USA). Sigma-Aldrich (Sigma-Aldrich, St. Louis, MO, USA) supplied AgNO_3_ (CAS NO. 7761880). The Kinetic Turbidity Assay (NCL Method STE-1.2) was used to test endotoxin concentration in four AgNP stock suspensions. The diameter of AgNPs was determined by transmission electron microscopy (TEM, JEOL 1010, Tokyo, Japan). The measurement of mass concentration and corresponding number concentration of the Ag in stock solutions was measured by Inductively Coupled Plasma Mass Spectrometry (ICP-MS, Thermo Fisher X Series 2, Waltham, MA, USA). We assumed a 5 nm thick layer of polymeric capping ligand (PEG, PVP, BPEI) on each particle and a 0.5 nm thick layer of molecular citrate for estimating the total mass of capping ligand in each solution. A UV-Visible Spectrophotometer (Agilent 8453, Agilent Technologies, Santa Clara, CA, USA) was used to investigate the spectral properties of materials. Zetasizer Nano ZS (Malvern Instruments Ltd., Malvern, UK) was used to measure the hydrodynamic diameter and zeta potential of nanoparticles.

### 2.2. Media Impact on Physicochemical Properties of AgNPs

To identify media impact on AgNPs’ physicochemical properties, Citrate AgNPs (1.14 mg/mL) were added to 1 mL of the different media, respectively. The media were deionized water, saline (0.85%, NaCl: 8.5 g/L), PBS (10× phosphate-buffered saline, NaCl: 80 g/L, KCl: 2 g/L, Na_2_HPO_4_: 14.4 g/L, and KH_2_PO_4_: 2.4 g/L, pH 7.4), serum (Fetal Bovine Serum), and α-MEM (Minimum Essential Media α with 10% FBS and 1% antibiotics). After shaking by hand for about half a minute, the samples were analyzed by TEM and dynamic light scattering (DLS).

### 2.3. Aging of AgNPs in 4 °C Storage

To identify the aging of AgNPs during storage, Citrate AgNPs, PEG AgNPs, PVP AgNPs, and BPEI AgNPs were kept for 1 year in a fridge at 4 °C. The aggregation, size, and shape were analyzed by TEM and DLS as described above. The dissolution of AgNPs was analyzed by ultracentrifugation (rpm using a Beckman 45Ti rotor, 30 min, Beckman Coulter Life Sciences, Brea, CA, USA) and ICP-MS. The toxicity of AgNPs to Hepa1c1c7 cell lines was analyzed before and after storage in the fridge.

### 2.4. Cell Lines and Cell Culture

The Department of Environmental Toxicology, University of California, Davis provided Hepa1c1c7 cells. They were maintained at 37 °C with a humidified atmosphere of 5% CO_2_ in an incubator. Cells were cultured in α-MEM (Gibco^®^ Minimum Essential Media α) with 10% FBS (Gibco^®^ Fetal Bovine Serum) and antibiotics.

### 2.5. Cell Viability

CellTiter-Glo^®^ luminescent assay (Promega, Madison, WS, USA) was used to measure cell viability. Detergents were added to the reagent to break cell membranes. This resulted in the release of ATPase inhibitors to media and their stabilization. ATPases are a group of enzymes that catalyze the hydrolysis of a phosphate bond in adenosine triphosphate (ATP) to form adenosine diphosphate (ADP). The assay is based on the conversion of beetle luciferin to oxyluciferin by a recombinant luciferase in the presence of ATP. The quantity of ATP in cells is proportional to the observed luminescence. The observed luminescence was measured by a microplate reader machine (Spark^®^ multimode microplate reader, Tecan Group Ltd. Männedorf, Switzerland). White, opaque, walled, 96-well plates (Nunc, Myriad industries, San Diego, CA, USA) were used for experiments. The ATP assay was conducted by platin, 5 × 10^3^ cells per well. Different concentrations (0, 0.1, 1, 5, 10, 25, and 50 µg/mL) of Citrate AgNPs, PVP AgNPs, PEG AgNPs, BPEI AgNPs, and AgNO_3_, for 24 h, were used for treatment.

### 2.6. Statistical Analysis

Statistical analyses for the cell viability were done using analysis of variance (ANOVA) and Tukey’s test with significance level at *p* < 0.05. Levene’s test was used to check homogeneity of variances. SPSS version 16 was used for the analyses.

## 3. Results

### 3.1. Characterization of AgNPs

All studied AgNPs were monodispersed with similar primary size (around 30 nm), as shown by TEM images (Figure 1A–D). ζ potential varied from −22.9 mV to +46.5 mV in the four types of AgNPs.

The stock suspensions were stable in Milli-Q water media, as evidenced by the hydrodynamic size of all AgNPs. Figure 1a–d presents concentrations of each capping ligand and content of endotoxin in stock suspensions.

### 3.2. Media Impact on Citrate AgNPs’ Physicochemical Properties

To identify the media impact on citrate AgNPs, we studied physicochemical properties of Citrate AgNPs in different media (Figure 2b–f). Our results show that media had a significant impact on the aggregation and zeta potential of citrate AgNPs. TEM images show that Citrate AgNPs are aggregated significantly in PBS and α-MEM (Figure 2D,F). The color of Citrate AgNPs changed from yellow to gray in PBS (Figure 2d). The zeta potential increased in serum and α-MEM but decreased in deionized water and saline (Table 1). The media impact on Citrate AgNPs was as follows: deionized water < saline < serum < α-MEM < PBS.

### 3.3. Transformation of AgNPs with Different Surface Coatings during Storage

Our aging studies show that the changes in physicochemical properties of AgNPs after one year relates to their surface coatings. The TEM images showed significant aggregation of Citrate AgNPs, and slight aggregation of PEG AgNPs and PVP AgNPs during one-year storage (Figure 3, Table 1). The zeta potential of all AgNPs is changed after one year (Table 1). However, the hydrodynamic size of all studied AgNPs did not show significant change after one year of storage. The dissolution of AgNPs was dependent on the coating of AgNPs. The BPEI AgNPs showed the highest dissolution (371.1 µg/L), and the citrate AgNPs showed the lowest dissolution (3.12 µg/L) (Table 1).

### 3.4. Dynamics of AgNPs’ Toxicity to Hepa1c1c7 Cells

Cell viability studies show higher toxicity of AgNPs stored for one year to Hepa1c1c7 cells as compared to particles tested without aging (Figure 4). The observed effect concentration of the initial AgNPs (Citrate AgNPs, PVP AgNPs, and BPEI AgNPs) to Hepa1c1c7 cells was 5 µg/L, which was five times higher than 1 µg/L observed for the aged AgNPs (Citrate AgNPs, PVP AgNPs, and BPEI AgNPs). The aged BPEI AgNPs also showed much higher toxicity than non-aged BPEI AgNPs (Figure 4). The EC_50_ of PEG AgNPs and BPEI AgNPs to Hepa1c1c7 cells decreased four and five times after one year of aging, respectively (Figure 4C).

## 4. Discussion

### 4.1. Media Impact on Citrate AgNPs’ Transformation

Our results showed that the media impact on Citrate AgNPs is lowest in deionized water, highest in PBS and Serum, and α-MEM is in between. We found that Citrate AgNPs aggregated significantly in 10× PBS, where the color changed from yellow to gray (Figure 2d,D). The gray color may result from the formation of AgCl by the Cl^−^ in PBS and dissolved Ag of Citrate AgNPs. Our results are consistent with others who conclude that silver ions are mainly precipitated as AgCl in the presence of chloride. Because phosphate is mostly protonated to hydrogen phosphate and dihydrogen phosphate in a moderate pH (around 7), silver phosphate was never detected [14]. We did not observe significant aggregation of Citrate AgNPs in saline (0.85%). The difference may be due to the lower concentration of Cl^−^ (8 g/L) in saline than the Cl^−^ (80g/L) in PBS. This result is consistent with the aggregation kinetics of the electrostatically stabilized AgNPs (e.g., citrate AgNPs). The ionic strength correlates with aggregation increases up to a certain ion concentration in the medium [15].

Nanomaterials can interact with proteins, peptides, and lipids (nanocorona) on their surface once the nanomaterials enter into biological systems [16,17]. The formation of a nanocorona depends on multiple characteristics of the biological media and the physicochemical properties of the nanomaterial [18,19,20,21]. For example, we found that the serum and α-MEM increased the hydrodynamic size of Citrate AgNPs and decreased the zeta potential of Citrate AgNPs, which increased the aggregation of Citrate AgNPs in media and may alter their bio-distribution, clearance, activity, and toxicity in organisms. By altering the zeta potential of Citrate AgNPs, the diversity of the proteins forming the protein corona may be affected, via changes in nonspecific electrostatic interactions with the proteins. Shannahan et al. (2013) found that that the number of amphiphilic proteins increases with the negativity of the AgNP’s zeta potential, indicating that electrostatic interactions play a major role in nanocorona formation [22].

Our results showed that citrate coating plays an important role in both AgNP stability and antibacterial activity. We found that the presence of salt (NaCl) increases the aggregation of Citrate AgNPs but the presence of serum decreases their aggregation in different media. Previous studies showed that the interaction of AgNPs with serum albumin had a significant effect on their antibacterial activity. Citrate AgNPs exhibited antibacterial properties due to minimized interactions with serum proteins [23]. However, uncapped AgNPs exhibited no antibacterial activity in the presence of serum proteins.

### 4.2. Impact of Aging on AgNPs’ Aggregation, Shape, Surface Charge, Size, and Dissolution

Interaction of NPs with the environment significantly changes their physical and chemical properties because the high surface to volume ratio and reactivity of NPs makes them highly dynamic in environmental systems [3]. However, AgNPs’ aging may not only show changes in environmental media, but also during sample storage.

#### 4.2.1. TEM Image Analysis for Aggregation and Shape of AgNPs

The TEM images showed that Citrate AgNPs had substantial aggregation resulting in increased size, and change in shape from spherical particles to triangular or irregular shapes. Such shape transformations have been found in other studies [24,25,26,27,28,29,30,31]. Citrate AgNPs are stabilized by charge repulsion where the citrate is weakly bound to the core Ag. The aggregation of Citrate AgNPs after one year of storage may be due to the decreasing charge repulsion with the changes of the ion strength in the medium.

There were no observable changes in size and shapes of PVP and PEG-coated AgNPs, but a slight aggregation of PVP and PEG-coated AgNPs were observed after one year of storage. The relative stability of PEG and PVP-coated AgNPs in comparison to Citrate AgNPs may be because PEG and PVP are sterically stabilized AgNPs, which are strongly bound to the core and potentially permeable to solutes and solvents.

BPEI AgNPs showed only slightly decreased size and aggregation after one year of storage. BPEI AgNPs are electrosterically stabilized and have abundant amine motifs that can form stable complexes with Ag atoms, which can explain their relative stability.

#### 4.2.2. Hydrodynamic Size and Zeta Potential of AgNPs by DLS Analysis

The DLS analysis did not show significant hydrodynamic size changes for any AgNPs in media after one year of storage. The findings of the TEM images analysis discussed above contradict these results. Because DLS provides information on the hydrodynamic radii of particles in solution, an increase in AgNPs’ size from DLS over TEM measurements is expected. It can indicate a successful capping of AgNPs with coatings [32]. Our results are not consistent with the results of others, maybe due to the analytical deficiencies in the DLS methodology we used. However, the results for the hydrodynamic size indicated that AgNPs in the medium were still stable after one year of storage. Citrate AgNPs and PVP AgNPs showed increased zeta potential but PEG AgNPs and BPEI AgNPs showed decreased zeta potential after one year of storage. The increased zeta potential of Citrate and PVP-coated AgNPs demonstrated a more stable dispersion of AgNPs than in the beginning. This is due to the degree of electrostatic repulsion between adjacent, similarly charged particles in a dispersion and can magnify the zeta potential of particles. A high zeta potential confirms stability for small molecules and particles. PEG AgNPs showed significantly decreased zeta potential, but the low molecular weight PEG can neutralize the surface charge and stabilize AgNPs through steric hindrance.

#### 4.2.3. Dissolution of AgNPs

The surface coating was found to be important for the dissolution of AgNPs based on the analysis by ICP-MS and ultracentrifugation. The positively charged BPEI AgNPs showed the highest dissolution (371.1 µg/L) but the negatively charged Citrate AgNPs, PEG AgNPs, and PVP AgNPs showed lower dissolution of 3.12 µg/L, 67.36 µg/L, and 130.60 µg/L, respectively (Table 1). NP surfaces can be covered imperfectly because the capping molecules can be very big for BPEI AgNPs. In the exposed areas, the AgNPs can be easily dissolved and oxidized. Another contribution to their faster dissolution is related to the ionic strength of positively charged BPEI. The AgNPs’ dispersions can be higher than in the case of the neutral or negatively stabilized particles. The capping molecules in the case of the negatively charged AgNPs are very small, resulting in a thin surface layer and their faster dissolution. This may be the reason why we also detected high ionic Ag in PVP AgNPs and not in Citrate AgNPs during storage. This is inconsistent with other studies, which showed that the dissolution of AgNPs decreased with increasing pH [33]. Therefore, the dissolution of AgNPs during storage under optimal conditions is relevant to the surface coatings.

### 4.3. Aging Impact on the Toxicity of AgNPs’ to Hepa1c1c7 Cells

We found that one-year storage significantly changes the physicochemical properties of AgNPs. The 24-h toxicity of AgNPs aged for one year to Hepa1c1c7 cells is higher than the toxicity of pristine AgNPs (Figure 4). The observed effect concentration of the initial AgNPs (citrate AgNPs, PVP AgNPs, and BPEI AgNPs) to Hepa1c1c7 cells was 5 µg/L, which was five times higher than the 1 µg/L of the aged AgNPs, including citrate AgNPs, PVP AgNPs, and BPEI AgNPs. The observed effect concentration of the aged PEG AgNPs to Hepa1c1c7 cells was increased 50 times (Figure 4). Aged BPEI AgNPs also showed much higher toxicity than the pristine BPEI AgNPs (Figure 4C). The increased toxicity of BPEI AgNPs may be due to the high amount of released ionic Ag inducing cell toxicity. The toxicity of aged Citrate AgNPs may be relevant to the aggregations of Citrate AgNPs. Our results are consistent with others showing the impact of aging on NP properties resulting in influences on their toxicity [24,25,26,27]. For example, Kittler et al. (2010) showed that silver NPs are increasingly toxic to human mesenchymal stem cells due to the increased release of silver ions [34].

## 5. Conclusions

Our study showed that media had a significant impact on the physicochemical properties of Citrate AgNPs. The electrolyte solution containing positive ions (e.g., Na^+^) can increase the ionic strength in media, which induced significant aggregation of Citrate AgNPs in PBS. However, Citrate AgNPs in saline did not show aggregation, which indicated that the kinetics of aggregation depends on the ionic composition of media. The decreased zeta potential of Citrate AgNPs in serum demonstrates that the diversity of the proteins forming the protein corona may be affected, reducing the aggregation of AgNPs. The aging of AgNPs with different coatings shows that the observed changes in AgNPs’ toxicity are related to different processes such as dissolution, agglomeration, and capping agent degradation. Our study confirms the strong impact of aging and media properties on increased toxicity of AgNPs. These processes need to be explicitly modeled to predict the fate of silver in the environment and humans.

## Figures and Tables

**Figure 1 nanomaterials-10-02255-f001:**
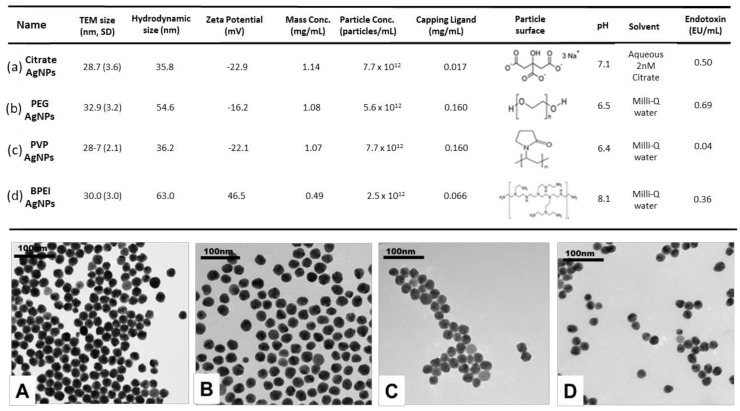
Characterization of silver nanoparticles (AgNPs). (**a**–**d**) Physicochemical properties of Citrate AgNPs, polyethylene glycol (PEG) AgNPs, polyvinylpyrrolidone (PVP) AgNPs, and branched polyethyleneimine (BPEI) AgNPs in stock suspensions. (**A**–**D**) TEM images of surface-modified AgNPs at 0 time. (**A**) Citrate AgNPs, (**B**) PEG AgNPs, (**C**) PVP AgNPs, and (**D**) BPEI AgNPs. The polydispersity index (PI) of DLS in each medium was less than 0.05.

**Figure 2 nanomaterials-10-02255-f002:**
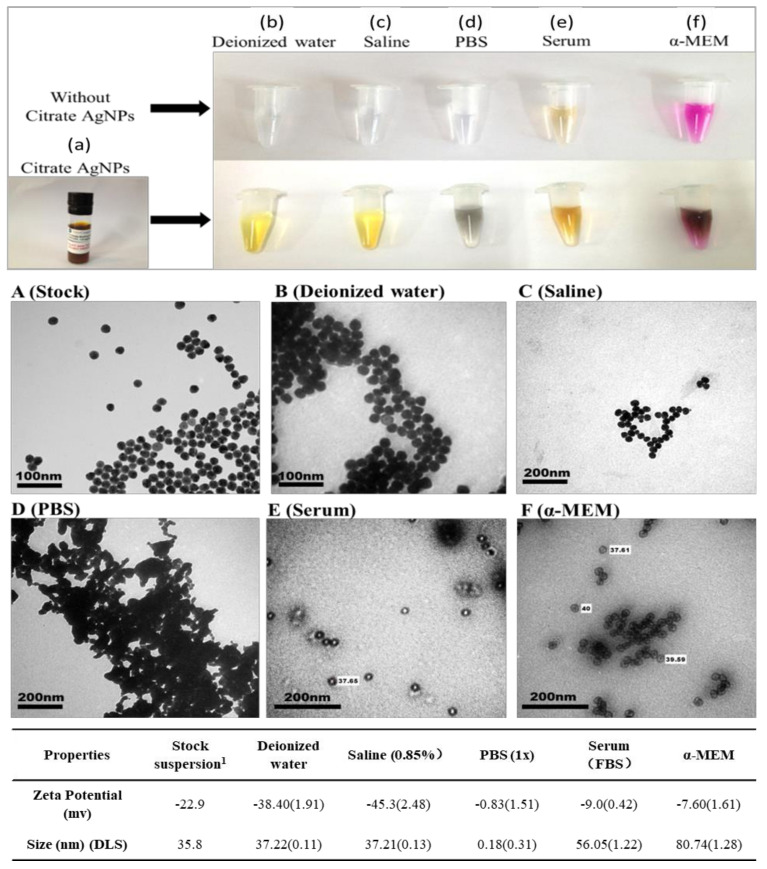
Characterization of Citrate AgNP suspension in different media. (**a**–**f**) Photo images of media and Citrate AgNPs in media. (**A**–**F**) TEM images of Citrate AgNPs in different media. The table at the bottom shows the zeta potential and size of DLS of Citrate AgNPs in different media. The PI of DLS in each medium was less than 0.05, except the PI of DLS in PBS medium. (Note 1: the stock suspension of Citrate AgNPs in aqueous 2 nm citrate).

**Figure 3 nanomaterials-10-02255-f003:**
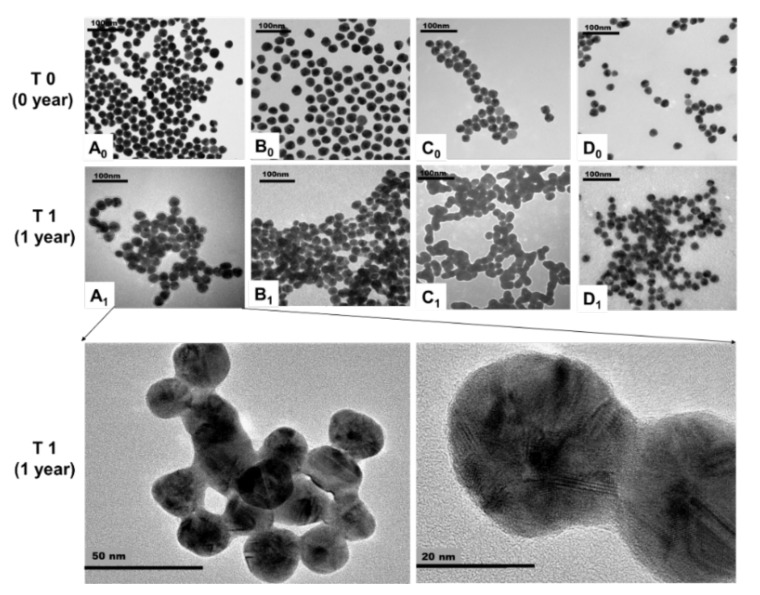
TEM images of surface-modified AgNPs. **A_0_**–**D_0_**: The initial images of Citrate AgNPs, PEG AgNPs, PVP AgNPs, and BPEI AgNPs. **A_1_**–**D_1_**: The one-year TEM images of Citrate AgNPs, PEG AgNPs, PVP AgNPs, and BPEI AgNPs in storage medium.

**Figure 4 nanomaterials-10-02255-f004:**
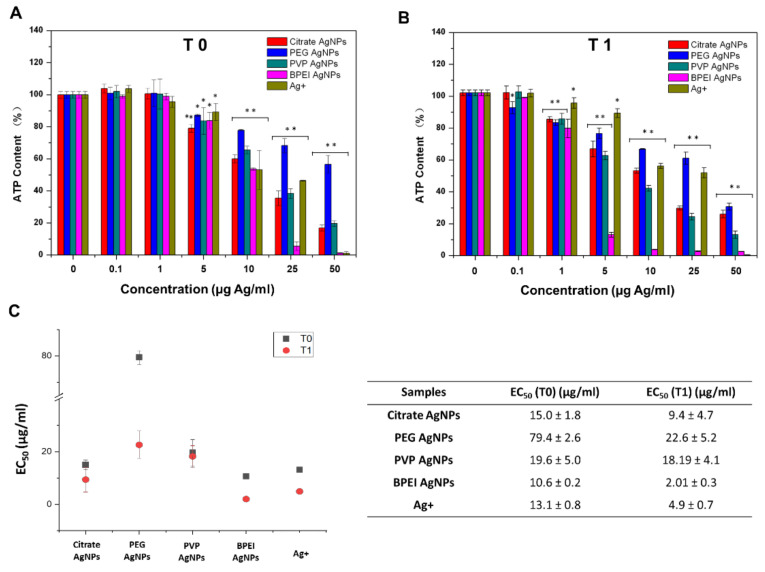
Dynamics of AgNPs’ toxicity to Hepa1c1c7 cells. Data obtained from luminescent assay for silver nanoparticles in Hepa1c1c7 cells 24 h. (**A**). The toxicity of the pristine AgNPs to Hepa1c1c7 cells. (**B**) The toxicity of stored one-year AgNPs in lab to Hepa1c1c7 cells. The *y*-axis represents the percent of reduction in ATP content compared to control. The *x*-axis represents the concentrations of silver nanoparticles. The different colors of the bars identify different silver nanoparticles (Citrate AgNPs, PEG AgNPs, PVP AgNPs, BPEI AgNPs, and Ag^+^). (**C**) EC_50_ of AgNPs to Hepa1c1c7 cells. The value represents the mean ± standard deviation of three replicates. The asterisks * *p* < 0.05, ** *p* < 0.001 comparing the untreated and treated (AgNPs, Ag^+^) cells.

**Table 1 nanomaterials-10-02255-t001:** Characterization of initial AgNPs and AgNPs stored for 1 year in a fridge (mean (SD), *n* = 3). The PI of DLS in each medium was less than 0.05.

Name	TEM Size (nm)		Hydrodynamic Size (nm)		Zeta Potential (mV)		Mass Conc. (mg/mL)		Released Ag^+^ (µg/L)	
**Time (Year)**	T0	T1	T0	T1	T0	T1	T0	T1	T0	T1
**Citrate AgNPs**	28.7(3.6)	76.2(34.0)	35.8	34.4	−22.9	−48.5	1.14 (0.06)	1.09 (0.02)	0.43 (0.10)	3.12 (0.14)
**PEG AgNPs**	32.9(3.2)	60.5(30.5)	54.6	55.8	−16.2	−2.71	1.08 (0.01)	1.01 (0.03)	0.45 (0.03)	67.36 (4.26)
**PVP AgNPs**	28.7(2.1)	31.3(34.9)	36.2	40.5	−22.1	−31.5	1.07 (0.03)	0.91 (0.03)	0.58 (0.10)	130.60 (5.40)
**BPEI AgNPs**	30.0(3.0)	26.6(4.3)	63.0	70.4	46.5	37	0.49 (0.03)	0.39 (0.02)	0.80 (0.07)	371.10 (14.82)

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
