# Peer review of "Transformation and Cytotoxicity of Surface-Modified Silver Nanoparticles Undergoing Long-Term Aging"

_nanomaterials, 2020, doi:10.3390/nano10112255_

Round 1
Reviewer 1 Report
in the introduction, the authors focused only on the Antimicrobial effect of AgNPs. they need to change the beginning of the introduction and add other applications of AgNPs (as for example, 10.1016/j.talanta.2020.121040). Indeed, the aggregation problem is not limited to the antimicrobial application.
Line 45 : remove bio/eco-corona
Line 111 : 5x103 (write properly)
The authors must add polydispersity index in all tables and DLS profiles in the supplementary information. Moreover, they must add a more detailed experimental part concerning the characterisation of size and zeta-potential by DLS.
Figure 4 : the authors must calculate EC50 and add a table with the values
Author Response
Responses to Reviewer 1 comments
in the introduction, the authors focused only on the Antimicrobial effect of AgNPs. they need to change the beginning of the introduction and add other applications of AgNPs (as for example, 10.1016/j.talanta.2020.121040). Indeed, the aggregation problem is not limited to the antimicrobial application.
Response: we agree the reviewer’s comments. The other applications of AgNPs in the beginning of introduction were added.
Line 45: remove bio/eco-corona
Response: The bio/eco-corona was removed.
Line 111: 5x103 (write properly)
Response: the number was revised.
The authors must add polydispersity index in all tables and DLS profiles in the supplementary information. Moreover, they must add a more detailed experimental part concerning the characterisation of size and zeta-potential by DLS.
Response: we are very sorry we didn’t keep the DLS profiles during the measurement. The polydispersity index (PI) of DLS in each media was less than 0.05 except the PI of DLS in PBS media. We added all these information to explain these results in text. Please see them in captions of table and figures.
Figure 4: the authors must calculate EC50 and add a table with the values.
Response: the EC 50 was added. Many thanks for the suggestions.

Reviewer 2 Report
The manuscript of Pang et al. entitled “Transformation and cytotoxicity of surface modified silver nanoparticles undergoing long-term aging” reports an interesting research on ensuring safety of surface modified silver nanoparticles (AgNPs). The Authors study the transformation of AgNPs in different media, characterize their transformation and check the toxicity these coated silver nanoparticles on Hepa1c1c7 cells after a long-term aging process.
The results achieved show that that storage and media of AgNPs influence their transformation and that the resulting changes in the AgNPs' physicochemical properties influence their toxicity.
Given the importance of this finding, the manuscript may be accepted for publication after some changes listed below.
1- the authors should add a new paragraph where they give information on the media used in the materials and methods paragraph or review 2.2 paragraph in material and methods. In this way, the composition of the PBS reported in the discussions (page 6 line 177) must be eliminated;
2- please add information on the ultracentrifuge and on the rotor used in paragraph 2.3;
3- what instrument was used to measure the luminescence?;
4- throughout the manuscript change “&” with “and”;
5- in the table the measurements shown do not have the standard deviation (±DS). Is this correct for all shown measures?;
Author Response
Response to Reviewer 1 comments
The manuscript of Pang et al. entitled “Transformation and cytotoxicity of surface modified silver nanoparticles undergoing long-term aging” reports an interesting research on ensuring safety of surface modified silver nanoparticles (AgNPs). The Authors study the transformation of AgNPs in different media, characterize their transformation and check the toxicity these coated silver nanoparticles on Hepa1c1c7 cells after a long-term aging process.
The results achieved show that that storage and media of AgNPs influence their transformation and that the resulting changes in the AgNPs' physicochemical properties influence their toxicity.
Given the importance of this finding, the manuscript may be accepted for publication after some changes listed below.
1- the authors should add a new paragraph where they give information on the media used in the materials and methods paragraph or review 2.2 paragraph in material and methods. In this way, the composition of the PBS reported in the discussions (page 6 line 177) must be eliminated;
Response: The information of the media was added in 2.2.
2- please add information on the ultracentrifuge and on the rotor used in paragraph 2.3;
Response: The information of the ultracentrifuge and the rotor was added in 2.3.
3- what instrument was used to measure the luminescence?;
Response: it was measured by the microplate reader (Spark®, TECAN). The information was added in 2.5, page 112.
4- throughout the manuscript change “&” with “and”;
Response: Many thanks. They are changed.
5- in the table the measurements shown do not have the standard deviation (±DS). Is this correct for all shown measures?
Response: the ISO standard for DLS suggests that the results should be reported using the mean and the polydispersity index (PI). Therefore, we didn’t give the SD for the DLS data.

Round 2
Reviewer 1 Report
line 37 : change products by applications
Author Response
Responses to the Reviewer 1 comments
line 37 : change products by applications
Response: Many thanks for the suggestions. The products was changed to applications.